# On the Tolerance of Children's Literature Criticism: Psychoanalysis, Neighborliness, and *Pooh*

**Neil Cocks**

Department of English Literature, Whiteknights Campus, The University of Reading, Reading RG6 6EL, UK; n.h.cocks@reading.ac.uk

**Abstract:** This article challenges David Rudd's recent criticism of 'The Reading Critics' school of children's literature criticism, which he takes to be problematic in so far as it is intolerant towards traditions that stray outside its own narrow concerns. Rudd forwards in its place an approach that is generous and dynamic. Through a close reading of Rudd's analysis of both *Winnie-the Pooh* and psychoanalysis, this article understands the politics and poetics of tolerance to open some difficult questions. What are the limits of tolerance? Is what Rudd forwards merely a tolerance of the tolerable? Is his forgiving attitude to the work of 'The Reading Critics', as he mourns their passing, tolerance also? What if these critics were to object to such tolerance, or read violence or erasure within it? Most significantly, this article is interested in how such tolerance, and the celebration of open community, fits within the 'broadly Lacanian framework' that Rudd elsewhere champions. As Lacan has, at best, an ambivalent attitude to the politics of neighborliness, this article argues that the defense of a 'broad' and tolerant approach to theory that calls upon his work is only made possible by arguments that neglect the specifics of Lacan's writing.

**Keywords:** *Winnie-the-Pooh*; David Rudd; Lacan; children's literature; neighborliness; Karin Lesnik-Oberstein; Jacqueline Rose; *jouissance*; tolerance; 'The Reading Critics'





## 1. Introduction

What follows is an oblique reply to David Rudd's recent article in *The Lion and the Unicorn* on 'The Reading Critics' of children's literature. The reply is necessary, in so far as Karin Lesnik-Oberstein, Sue Walsh and myself are described in this work as engaging in 'a particularly futile form of theory war' (Rudd 2020a, p. 96). Although Rudd makes clear he wishes to move away from some of the more unpleasant language utilised against us, my sense is that a figuring of our work as both sterile and surpassed should expect—and, indeed, probably wants—a response. The reply is oblique because we have been here before. As Rudd acknowledges, when 'The Reading Critics' and certain other children's literature scholars meet, the result is an analysis that is interminable.[1] My aim in what follows, therefore, is to shift the terms of debate.

The disagreement upon which Rudd's article turns will be introduced in more detail below, but it is centrally concerned with the extent to which the lived experience of children can be understood to transcend framing discourses. On the one side, critics such as Perry Nodelman (2008), Marah Gubar (2010) and Karen Coats (2013) claim that, whilst critics should attend to the constructed and historical nature of childhood, there is a danger in downplaying the voices and experiences of actual young people, as this can lead to silencing, and a partial understanding of real-world effects. Gubar offers the following formulation to this effect:

> As David Rudd and I have argued, such accounts—which often claim not to be saying anything about children—actually suggest that adults have power, voice, and agency and children do not [ . . . ] Such discourse is deeply problematic, precisely because of the point about self-fulfilling prophecies that these critics

themselves have made [ . . . ] If we take that point seriously (which I do), the mere act of describing young people as voiceless can itself help render them voiceless. (Gubar 2013, p. 452)

In opposition to this, critics such as Karin Lesnik-Oberstein (1994) and Jacqueline Rose (1984) argue for the necessity of reading in detail the perspectives and investments through which childhood is constituted, suggesting that this will problematise notions of child voice and identity upon which critics such as Gubar depend. In his article, Rudd builds on previous critiques of Lesnik-Oberstein and, to a lesser extent, Rose, whilst also suggesting the central fault of 'The Reading Critics' is the dogmatism with which they reject any appeal to the real of childhood.[2]

My response aims to break out of the 'vs' binary, firstly by framing the meeting of Rudd and 'The Reading Critics' in terms of wider arguments concerning what can, cannot, should, and should not be tolerated, arguments that, I would contend, are at the core of any consideration of political existence.[3] More specifically, I will be questioning Rudd's celebration of an approach to children's literature that is tolerant of opinions from which it diverges and resistant to iron dualisms and stalemates. For Rudd, we should be seeking 'hieratic' approaches to theory that are 'healthily eclectic and tolerant while avoiding calcification' (Rudd 2020a, pp. 101–2).

I certainly take this celebration of tolerance to open some difficult political questions. What, I would ask, are the limits of tolerance? Is what Rudd forwards merely a tolerance of the tolerable? Is his *forgiving* attitude to the work of 'The Reading Critics', as he mourns our passing, tolerance also? What if we were to object to such tolerance, or read violence or erasure within it? Most significantly, for this article, at least, I am puzzled as to how such tolerance, and the celebration of an open (non-interpretive) community, fits within the 'broadly Lacanian framework' that Rudd elsewhere champions (Rudd 2013, p. 80). It all sounds very *neighbourly*. Well, perhaps not, although, as I shall argue, whichever way this goes, it ends in a rather dark place.

In stepping to the side of the antagonistic structure within which (I would contend) Rudd remains, and his broad approach to theory, both of which are set up in *The Lion and the Unicorn* article as the frame for any potential reply, I will begin by responding in detail to an alternative area of Rudd's research, his readings of *Winnie-the-Pooh*, especially that forwarded in *Reading the Child in Children's Literature: An Heretical Approach* (2013). Rudd's text is chosen because it has been widely praised (Gubar 2013; Thacker 2014; Eisenstadt 2014) is taken to be exemplary in its reading of Lacan (McGillis 2016), and continues to attract praise from scholars both inside and outside Anglo-Saxon Children's Literature studies (García-González 2022; Lassén-Seger 2015). Milne's text is chosen because it is one Rudd frames in terms of discussions of Lacanian psychoanalysis (Rudd 2013) and with which he remains interested (Rudd 2021). Most importantly for this present article, Rudd understands *Winnie-the-Pooh* to be a tolerant text that benefits from a tolerant and 'dynamic' reading. Through a detailed analysis of both Milne's text and Rudd's response, I will suggest some of the difficulties with the tolerance the latter champions and relate this to the freedom and eclecticism understood to be enabled by a 'broadly Lacanian' approach (Rudd 2013).

Such an analysis will open up a further discussion of Rudd as a reader of Lacan, this centering on the appeal to Lacan in the analysis of *Winnie-the-Pooh* introduced above, and also Rudd's chapter on 'Psychoanalytic Approaches to Children's Literature' in the popular academic text-book *Modern Children's Literature: An Introduction* (2014). Here I identify and oppose what I see as Rudd's generalized approach to theory through focusing at some length on just one of his Lacanian ideas, central to the reading undertaken in the previous section: *jouissance*. I choose *jouissance* above anything else because it allows me to further question discourses of tolerance. I argue that, for Lacan, the term refers not only to the excess of pleasure required by Rudd's argument, but is dependent also on censure and limitation. This enables a response to Rudd's claims to work within 'a Lacanian framework',

whilst also figuring the tolerance he advocates as problematically requiring a mastery of both its own position and the permissible that is tolerated.

In the penultimate section of the article, I take up in more detail Rudd's dismissal of the psychoanalytic tradition in children's literature exemplified for him by Jacqueline Rose, a dismissal that has become orthodoxy within Anglo-Saxon children's literature criticism, and increasingly has a foothold within wider fields.[4]

I conclude by returning to a text by Lacan to do what Rudd does not: read psychoanalytic theory in detail and in perspective. The text—the meditation on St Martin from *The Ethics of Psychoanalysis*—is chosen for two reasons. Firstly, it is a text that Rudd, Coats, and other psychoanalytically inclined children's literature critics have not, and, I suggest, will not, engage, as within it Lacan sets out a critique of neighborliness that would be for them, I think, intolerable, one that sees political notions of tolerance and community subverted. Lacan is interested in the annihilating, asocial forces that such tolerance calls upon. I am interested in turn in how such forces might problematize Rudd's vision for the future of children's literature, in which a closed community of critics respectfully trade views and differences, having either wholly assimilated or expelled any group such as 'The Reading Critics' who would bring into question the values that enable the perpetuation of unthreatening exchange.

There is, however, a further and far simpler reason why a discussion of St Martin is unlikely to appear in conventional children's literature criticism: detailed and questioning readings of Lacan are not part of its practice. As suggested previously, such criticism, in my understanding, gestures towards a theoretical approach or framework without a sustained and text-focused analysis of it, this in ways that would not always be tolerated in other areas of humanities studies. The stakes are thus higher than a parochial debate amongst a small group of children's literature critics: what approaches are foreclosed within contemporary literary studies when founded on liberal tolerance? What problems attend any eclectic approach to an established body theory? In working through these issues, my hope is also to initiate a questioning of the limits on reading with contemporary psychoanalytic study, and the limits on the notion of identity (such as 'child') as construction within wider literary and social theory.

## 2. '[ . . . ] Playing with the Conventions of Language'

The last few years have seen a significant growth in published work on *Winnie-the Pooh*, with critics engaging a variety of approaches (post-colonial, post-humanist, historicizing psychoanalysis), and sometimes, as with Rudd's own work, calling upon more than one. Although often insightful and productive, I share Rudd's interest in the extent to which such interventions might be understood to avoid questions of reading: *Reading the Child in Children's Literature: An Heretical Approach* (2013) is significant in the way it foregrounds questions of narration in *Winnie-the-Pooh* in a way certain overtly 'political' critics arguably do not.[5] Although working within a 'Lacanian framework', Rudd is keen to promote 'energetics' over 'mechanics' and is sceptical of generalised or formulaic criticism: Rudd despairs of 'a certain predictability in much that appears in our journals as yet another feminist, post-colonial, eco-critical or "fill the gap" reading is undertaken' (Rudd 2013, p. 3). Take, for example, the work of Daphne Kuzter:

> We might respond to [Kutzer's reading of *Winnie-the-Pooh* as a colonial text featuring a disempowered child] by arguing that Milne's narrative technique is actually quite ingenuous in that he makes it plain that it is an adult telling these tales. Thus, in response to Kutzer's claim that 'Christopher Robin has been colonized by the adult narrator. He is not free to tell his own stories or to have a starring role in them', we might ask in what sense this storybook character could be more liberated: by having the stories feature Christopher Robin more prominently, by having him narrate in the first person? Wouldn't this, in fact, make the fictions more of a 'soliciting . . . chase, or seduction,' as [Jacqueline] Rose

terms it, concealing the adult writer—[Perry] Nodelman's 'hidden adult'—from the child reader? (Rudd 2013, pp. 65–66)

I am sympathetic to the force of this argument, centring as it does on the difficulty Kutzer, or any thematically inclined critic, faces in establishing a children's literature that remains wholly on the right side of colonial politics.[6] If this means that, for Rudd's analysis, there will always be a limit to liberation, *Winnie-the-Pooh* is figured as the most liberated of texts for children: Milne's text may not be unambiguously good, in other words, but it is as good as it gets. A willingness to counter the prevailing reluctance to read the language of *Winnie-the-Pooh* would seem essential for this assessment, as Milne is understood to arrive at the maximization of possible liberation through 'ingenious' 'narrative technique'. Indeed, the claim for the text is that 'it is very complex, and certainly confounds any notions of language transparently telling the story', such notions apparently being attributable to Kutzer (Rudd 2013, p. 66).

A repeated argument within Rudd's engagement with *Winnie-the-Pooh* is that this questioning account of language's representational function can be read in the text's problematization of names. This leads to its engagement with the opening lines of the 'INTRODUCTION':

> If you happen to have read another book about Christopher Robin, you may remember that he once had a swan (or the swan had Christopher Robin, I don't know which) and that he used to call this swan Pooh. That was a long time ago, and when we said good-bye, we took the name with us, as we didn't think the swan would want it any more. Well, when Edward Bear said that he would like an exciting name all to himself, Christopher Robin said at once, without stopping to think, that he was Winnie-the-Pooh. And he was. So, as I have explained the Pooh part, I will now explain the rest of it. (Milne [1926] 1973, no page number)

Rudd responds with the following:

> Apart from the confusion of Pooh living under this other name (Sanders), he is also simply 'Edward Bear' at the outset, who, we are informed, adopted the name of a swan: 'we took the name with us as we didn't think the swan would want it any more', playing on the standard notion of 'taking' someone's name. (Rudd 2013, p. 67)

It is here, as the analysis addresses a specific textual formulation, that a commitment to attentive reading can be read to stall. In Rudd's understanding, the quotation concerns 'Edward Bear' who 'adopted' the name of 'a swan', yet 'Edward Bear' is not, in my reading of the 'INTRODUCTION', implicated in the taking of the name: 'Edward Bear' says he would like a name 'all to himself', and Christopher Robin says he was Winnie-the-Pooh (and he was), sometime after the 'taking' has taken place. Even if, for some reason, we were to include 'Edward Bear' in the 'we', challenges remain, as this is a problematically collective 'we', rather than any singular subject, that 'take[s]', rather than 'adopt[s]', the name of 'the', not 'a', swan. Within the passage, the narrative perspective is not simply that of the collective 'we', but an 'I' which is a part of, yet exceeds, it. These are subtle points, but I would suggest significant, in so far as, for Rudd, the transparency of language is threatened by the claimed 'adoption' of a name by 'Edward Bear', rather than the various slippages I read to problematise this narrative. In conclusion, Rudd states that:

> Without elaborating further, the point should be clear that Milne is playing with the conventions of language [ . . . ] showing language's slippage in the way that these things do not stay in place ([ . . . ] confounding [Jacqueline] Rose's claim about the transparency of language in children's literature) (Rudd 2013, p. 67).[7]

Within this argument there is again a limit imposed upon on the 'slippage' that is being championed, the sense in which 'these things' [the names] do not stay in place. As this is 'clearl[y]' a matter of 'Milne' 'playing with the conventions of language', there can be no doubt as to what the author of *Winnie-the-Pooh* is doing. He is 'playing' and is so doing with a set of known and stabilised 'conventions'. Play is other, or subsequent, to the

conventional, it would seem, although the one can act upon the other. That there can be no uncertainty about this is confirmed in the way that the 'play' results in 'things' not staying in place and this amounts to 'showing language's slippage'. Thus, although it is claimed that the psychoanalytic critic Jacqueline Rose is wrong in what are taken to be her ideas concerning the transparency of language in children's literature, there can be no doubt as to what *Winnie-the-Pooh* 'show[s]': the certain truth of slippage. For Rudd, this suggests that:

> Rather than being the all-powerful coloniser [ . . . ] Milne (or the narrator) appears to be remarkably 'uneasy and tentative,' as [Barbara] Wall expresses it. For me, then, Milne seems to be troubled by this relationship, unsure as to where he stands in relation to the child and to language. (Rudd 2013, p. 67)

This questioning of the relationship between Milne, the child, and language is taken to be indicative of a lack of certainty that runs counter to the operation of hierarchical control: "Milne (or the narrator) appears to be remarkably "uneasy and tentative'" and 'I would suggest that, rather than narrational control, the beginning of Milne's text expresses the very opposite' (Rudd 2013, p. 66). In this formulation, the loss of narrational control does not necessitate the relinquishing of the author. The author 'appears' to be 'uneasy', and as such, it is argued, is not to be implicated in the kind of colonial discourse critiqued by Kutzer. I take Rudd to be implying that colonialism is derived from, and is expressed as, deliberate and unwavering control. The claim, as I read it, is that power can only be exercised by an 'all powerful' that correctly recognises itself as such, with the implication that confusion, self-doubt or play offer no potential threat. This is, I would contend, a politically dangerous assertion, and my first appeal in response would be to a history of colonial practice as, in part, piecemeal, improvised, anxious, and confused.[8] A further implication, I think, is that, in being limited to 'appear[ance]' in his reading of Milne, Rudd can stake a claim to the narrator's own status as 'uneasy and tentative'.

As *Reading the Child* is alive to methodological considerations, it has its own take on the implications of its approach to *Winnie-the-Pooh*:

> In the above reading I have pressed for a more open-ended approach to criticism, one that does not wish to deny others, or only so when they seem to shut down interpretation and, thereby, delimit the pleasure, or energetics of reading, which is why I have suggested this more 'heretical' approach. Though I have adopted a broadly Lacanian framework in my own interpretations, in that I have argued that such an approach helps open up a text, I am also guided by Lacan's attention to 'the letter of the text' (Fink 2004), where meanings undoubtedly arise and multiply, but always from the ground up, rather than being imposed from above, in the process, the *jouissance* of a text can be released, as the censorious monologic of the Symbolic is defied. (Rudd 2013, pp. 79–80)

It is the 'broad' status of the psychoanalysis that defines the 'heretical' approach, this 'open[ing] up a text'. The 'wish' of the reading is to overcome denial, yet despite encouraging 'other' readings, there is a resistance to others that 'seem to shut down interpretation', and to the 'monologic of the Symbolic'.[9] There is also a requirement to pay '"attention to the letter of the text'", which is understood to enable meaning to 'arise' without imposition or censor. Because the letter of the text is the location for the arising of meaning, the suggestion is that it originates there: in that sense, there is no other to the text. In the first instance, the acceptance of otherness is dependent upon its lack of threat, its failure to offer resistance: an otherness that demands nothing, one that is not, in this sense, other. In the second, meaning is prior to and limited by the censorious Symbolic, yet located nonetheless at the point of 'the letter'.

Are we to read in Rudd's appeal to this letter a Lacanian resistance to interpretation, the notion that there is something in textuality that cannot be made to signify?[10] Well, maybe, but Rudd's argument seems to me at every other point to work against such a claim. I take the appeal here instead to be to attentive reading, with this understood in terms of release, defiance, multiplicity, pleasure without delimitation, liberation from the imposition

of others. I do not read any engagement with the ironies that might be understood to attend such liberation. Thus, for example, although there is no acknowledgment at this stage of pleasure in repression, I take Rudd's argument to require such repression to be enacted in order that it might be avoided, with this enabling the release of pleasure.

It is helpful, at this juncture, to turn again to the reading of *Winnie-the-Pooh* forwarded within *Reading the Child*. A connection can be clarified between the anti-colonial position that Rudd's text grants Milne, and the liberalism I understand it to be promoting. Just as it is claimed that Milne cannot be authoritarian because he is tentative, playful, and confused, so Rudd's preferred reading approach must be on the side of liberation because it follows no set plan and is concerned with '*jouissance*'. The difficulty here is that it can be argued that it is precisely through such ad hoc qualities that the narration of *Winnie-the-Pooh* justifies the taking of a name of another. The 'INTRODUCTION' begins with a certain generosity to the other in the acknowledging of the limits of narrational power ('If you happen to have read another book about Christopher Robin, you may remember that he once had a swan [ . . . ]'), the notion that 'you' have an independence from the narration, with this linked to what Rudd figures as an 'easy and tentative' lack of firm 'control'. This acceptance of limitation is repeated in the claim in the brackets that 'I don't know which' party had ownership. The generosity is not sustained, however. Thus, in my reading, the uncertainty as to the fact of ownership is not about a symmetrical claim. Indeed, and as briefly introduced above, what is outside the brackets is not precisely repeated within them, as, for example, there is a change in article: whereas Christopher Robin owns 'a' swan, singular yet generic, it is 'the' swan, certain and differentiated, that might own Christopher Robin. To be owned is not the same as to own, it would seem, this problematising the notion that, despite ownership having occurred, the identity of the owned is unknown, and thus irrelevant.[11] The identity of 'a swan' can be understood to be further questioned, as after the brackets the narration claims Christopher Robin 'used to call this swan Pooh'. The/a swan is not the name of 'this swan', a name here being other than what a thing is. The subsequent clause, however, constructs the name in slightly different terms: what can be 'taken' by a 'we', rather than something the 'he' calls 'this swan'. I would contend that the lack of knowledge here on the part of the narration does not signify acceptance of or tolerance towards the difference of the other, as it perhaps did initially, so much as a heartless justification for theft. Because it is known that we were thinking that we did not know what this other was thinking, it can be divested of the object that is its name.[12]

This reading could be significantly extended, of course, but here I will bring this section to a close with two brief conclusions. Firstly, that Rudd's liberatory account of reading must at one stage call upon the repression it sets out to overcome. Secondly, that a return to the text of the 'INTRODUCTION' suggests that the generosity of Milne as understood by *Reading the Child* is not entirely free from what Kutzer defines as the colonial.

## 3. '[ . . . ] The Censorious Monologic of the Symbolic Is Defied [ . . . ]'

In this section, I will take up the first of these concluding points by asking: what are we to make of Rudd's account of *Winnie-the-Pooh* as a reading of, or commentary on, Lacan? Even without the challenge of returning to the question of the Lacanian letter, a movement we will enact by way of conclusion, Rudd can be read to offer what might be termed an uncanny psychoanalysis; it resembles Lacan, perhaps, but only in a way that brings home a certain lack of fit. At every turn, I'm afraid, I see a problem, so I will have to limit myself to just one: *jouissance*.[13]

My sense of the passage above is that it figures *jouissance* as synonymous with pleasure. Admittedly, in the glossary supplied in *Reading the Child*, Rudd does write of a 'surplus pleasure', 'unbearable', and 'heretical', and elsewhere he describes 'blissful, almost painful, joy' (Rudd 2013, pp. 47, 193).[14] In the quotation above, however, the claim is that we must guard against the delimitation of pleasure, that is, the restriction on Rudd's own 'dynamic' reading, and defying the 'censorious monologic' is surely to be taken as a repetition of this move: *jouissance* is a matter of abundance, freedom, and its realisation is to be encouraged.

Well, *jouissance* is certainly *excessive*, yet, as Glyn Daly states, it does not, indeed, 'equate simply to pleasure' (Daly 2014, p. 80). Pleasure, to follow the standard Lacanian line, is caught up in an economy of balance—the pleasure principle—and this requires prohibition: if everything is to achieve a state of equilibrium, a limit must be placed on pleasure. There is, it can be claimed, an illusion at work here, however, as the operation makes us think that if somehow censoriousness were to be removed, *jouissance* might, however dangerously, be achieved. *Jouissance* can thus come to be understood as a transgression of a prohibition, and, in the economy of desire, where the 'fundamental fantasy' forever holds out the neurotic hope of completion, it is what we constantly and futilely seek (Lacan [1966–1967] 2002; see also Turner 2017; Žižek 1997).

One upshot of this is that although ideology certainly relies on a renunciation of *jouissance*, we should not think of it as *prior* to repression. *Jouissance* is to be understood in terms of the limit it exceeds (see especially Lacan 1989. See also Lacan [1961] 2006, 2018; Žižek 2008; Zupančič 2019), it is 'the excess of pleasure produced by "repression" itself, which is why we lose it if we abolish repression' (Žižek 2013, p. 308). We must be very careful, then, in reading Rudd's claim that 'the *jouissance* of a text can be released, as the censorious monologic of the Symbolic is defied.' This is only the case if we see the release failing at the very point of success: the release occurs in so far as the act of defying takes place. It has no wholly autonomous existence beyond this. *Something* must be defied. There must be a limit. The oceanic sense of *jouissance* Rudd employs should thus be treated with caution. Censorious is required for *jouissance*, and its release never has the luxury of breaking free from the scene of breaking.

In relation to this, we might think about the opposition Rudd forwards between *jouissance* and 'the censorious monologic of the symbolic'. Here the appeal is, I suppose, to 'The Subversion of the Subject and the Dialectic of Desire in the Freudian Unconscious', as from this text we can indeed understand the censorious nature of the Symbolic (Lacan [1961] 2006). In Rudd's argument in the above quotation, however, there is no sense that I can see in which *jouissance* returns *as* 'the censorious monologic of the Symbolic'. For Lacan, as drive moves round its unchanging circuit, thus carving out a space of nothing that is set up retrospectively as its cause, it fails to capture a lost *jouissance* that was always anyway lost (Lacan [1961] 2006; Žižek 2008; Copjec 2015). In this very movement, however, there is, ironically, *jouissance*: precisely a surplus pleasure in repetition, a deathly pleasure. This, then, is part of the strangeness of *jouissance*. It can be achieved through fixity, from the liberation from the other; an affront to, rather a fulfilment of, meaning; excess rather than abundance.

In *The Ticklish Subject*, a work dedicated to refiguring Lacanian psychoanalysis in terms of a constitutive absence in the human subject, Slavoj Žižek brings together both the problem of *jouissance* returning as death drive and the 'paradox of jouissance [ . . . ] of *das Ding* which can be experienced only in a negative way—whose contours can be discerned only negatively, as the contours of an invisible void' (Žižek 2008, p. 44):

> Desire desperately strives to achieve *jouissance*, its ultimate object which forever eludes it; while drive, on the contrary, involves the opposite impossibility—not the impossibility of attaining *jouissance*, but the impossibility of getting rid of it. The lesson of drive is that we are condemned to *jouissance*: whatever we do, *jouissance* will stick to it; we shall never get rid of it; even in our most thorough endeavour to renounce it, it will contaminate the very effort to get rid of it (like the ascetic who perversely enjoys flagellating himself). (Žižek 2008, p. 355)

Releasing *jouissance*? A comforting dream. The moment of liberation will always see a return, but rather than restricting the subject, this can suggest an awful freedom; a recognition of that part of the subject that *insists*, that is not subject to imaginary relations. What I am gesturing towards here is an additional, although far from final, turn in Lacan's notion of *jouissance*: the nothing that escapes the symbolic, what the subject moves towards in *jouissance*, is also the nothing that lies strangely at the heart of the subject, a nothing that sets up an uncanny relationship, to which we will return in the conclusion to this

article when we think about what Lacan has to say about neighbourliness, and thus also the politics of tolerance.

This entirely cursory introduction to *jouissance* is not there to fill the gaps in Rudd's account: I am not a Lacanian and have no interest in forwarding a 'correct' Lacan; even if that were so, what I write here is far too full of holes, and, in any case, we are still waiting to return to the text of Lacan. Rather, it is to give some sense of what is problematic in just one aspect of Rudd's 'Lacanian framework'. It is a framework that can be read to have little interest in the constitutive tensions, the impossible returns and reversals, that mark Lacanian thought, with *jouissance* figured simply as (sometimes painfully intense) literary pleasure, multiplicity, and meaning that it is good to set free. In what sense is such framework 'Lacanian'?

### 4. '[ . . . ] Reading the Wrong Freud to Children'

It is my contention that it is the very 'heretical' approach—the freedom loving response that repeats the tolerance and generous confusion that it locates in the narration of *Winnie-the-Pooh*—that enables the privileging of a psychoanalysis that does not have to engage, let alone work through, the tensions within, or the specifics of, certain aspects of the Lacanian project. For Rudd, I think, to adhere too closely to any branch of psychoanalysis would run the risk of being caught, like 'The Reading Critics', in the 'monologic of the symbolic'. In the introductory chapter on 'Psychoanalytic Approaches to Children's Literature' in *Modern Children's Literature: An Introduction*, for example, Rudd reads *The Wind in the Willows* through Bettelheim, Klein, Jung, and Lacan, supporting an idea that all approaches should be at the disposal of the liberated, 'heretical' reader (Rudd 2014, pp. 50–51). This is something other than what might be figured as the heretical approach (via Hegel) of Žižek to Lacan, who would see himself as rigorously Lacanian.[15] Instead, I take Rudd's approach to be rooted in *whim*. Rudd would no doubt counter that he is just following the text, but that would suggest a text that demands to be met by a Lacanian 'framework' that skirts the demands of *jouissance*. Is *Winnie-the-Pooh* really such a text?

Although the 'framework' must be lacking, then, if it is to allow the reader to be 'nimble', Rudd, in his recent article for *The Lion and the Unicorn*, has supreme *confidence* in it, confidence enough to suggest to Jacqueline Rose, arguably the UKs leading Lacanian scholar, how she might up her game:

> What is perhaps most curious about Rose's case, though, especially given her involvement with Lacanian psychoanalysis [ . . . ], is how muted is this thinker's presence in her book on children's fiction. Only once, in a footnote, is Lacan overtly named [ . . . ]—not even receiving an index entry—despite the fact that his concepts underpin Rose's repeated cry, 'We have been reading the wrong Freud to children'. (Rudd 2020a, pp. 12–13)

What does it mean to read a better Freud to children in this sense? We can gain greater clarity as to Rudd's understanding through a further quotation: "[Karin] Lesnik-Oberstein does not follow Rose's more linguistic (Lacanian) reading of Freud but instead draws on, arguably, 'the wrong Freud' of D. W. Winnicott and others" (Rudd 2020a, p. 98). Rudd's Lacan is linguistic, and Winnicott is concerned with object relations, and it seemingly stands to reason that reading the former is the way to go. But what if one were to read Winnicott attentively and radically? Would it be possible to read Winnicott to children in a way that worked through questions of repetition, repression, framing, difference, and the constitutive gap necessary to analysis? A *psychoanalytic* reading of Winnicott? Might that be something other than the wrong Freud? Likewise, however linguistic Lacan's approach might be, is it not possible to read him in a way that works against this? I would contend, indeed, that if it is a 'fact that his concepts underpin Rose's repeated cry', then this is a Lacan who works against the linguistic, at least in one sense: a conceptual Lacan, where the concept is a solid foundation to something that is other than writing.

Let us turn again to Rudd's reading of Lacan, to confirm the kind of approach that is being forwarded. For Rudd, 'the poor substitute', language:

[ . . . ] drives humans, as we strive to overcome the sense of lack we experience, desiring to re-attain the Eden we think we once inhabited. Images of wholeness therefore beguile us, which is what advertising campaigns trade on ('Buy X and you too could be like this!') and, of course, in language we try to articulate our desires, moving from one signifier to the next, forever trying to repair our sense of incompleteness. This is what literary works temporarily proffer. For example, Anthony Browne's *Willy the Wimp* (1984) 'buys' into just such an imaginary notion of masculinity, demonstrating how we are strung between these three separate realms of existence: Willy's construction in the Symbolic as a 'wimp', as opposed to his Imaginary sense of being an alpha-male, a superhero; and then his undoing in the Real as he collides with a lamppost. (Rudd 2014, p. 47)

There are claims here that certainly might contribute to a Lacanian framework: Rudd understands the lost sense of wholeness to be illusory, there is a clear idea of the 'metonymy of desire', and an appeal to the 'extimate' nature of literature. This last opens up difficulties, however. Literature is read as a 'demonstration', as this is what *Willy the Wimp* achieves. Or perhaps not: despite the italics, *Willy the Wimp* is, I think, the character in this sentence—it is he, rather than the text as a whole, who buys into imaginary notions. However understood, this demonstration seems to me at odds with what literary texts 'proffer': the move from one signifier to the next, as we forever try to repair our sense of incompleteness. Is the demonstration incomplete or temporary, in this way? Does it fail? It would seem not, and thus literature has not proffered what it should. I read here not simply 'notions of language transparently telling the story', but literature as a clarifying, pedagogical *showing*. And what is made clear is merely that we are strung between words, daydreams, and bathetic material encounters.

I take what is introduced at this stage to be an analogical or a conventionally 'symbolic' approach to psychoanalytic reading. It is one to which Rudd has frequent recourse, not only in the appeals to 'long white phallic hands' and the like that repeat in his work, but in the broader sense in which children's literature texts are understood precisely to *demonstrate* Lacanian theory (Rudd 2008).[16] Despite the claim that children's literature is 'ineluctably haunted by the uncanny effects of language', a Lacanian reading is one that simply recognises the 'broadly' Lacanian narrative in such texts, one that testifies to, rather than works through, the instability of language (Rudd 2013, p. 119). It is an approach that is often repeated in children's literature criticism, most noticeably, perhaps, in the work of Karen Coats, Rudd's fellow advocate of Lacan-inflected and tolerant approaches (Coats 2007). Here we might introduce a slightly extended quotation from Jacqueline Rose, referenced in part by Rudd above: 'this is the most [ . . . ] prevalent' reading of Freud, and 'the form of interpretation—where one thing straightforwardly equals another—which seems to predominate in the analysis of Children's writing [ . . . ]', but it is also "the 'worst' of Freud", in so far as it 'by-passes any problem of language' (Rose 1984, p. 19).

This is not the only occasion in his *Lion and the Unicorn* article where Rudd offers a strangely partial quotation from Rose. It is Rudd's contention that the central difference between Rose and 'The Reading Critics' is that the former is explicitly committed to a child of the real: "Rose can quite legitimately distinguish between 'the child inside the book' and the one 'outside' that 'does not come so easily within its grasp'" (Rudd 2020a, p. 95). According to this argument, Rose, along with Jacques Derrida, has a belief in the real as opposed to construction.[17] To avoid confusion, then, the non-discursive child of the real is not, for Rudd, that Lacanian little piece of the Real, the object *a*, the object cause of our desires, but something more prosaic: a referent; an 'underlying' *reality* to which interpretation responds (Rudd 2020a, p. 98). It is not, to return to Žižek's previous formulation, what 'can be experienced only in a negative way—whose contours can be discerned only negatively, as the contours of an invisible void', but the actuality of 'referents that are extra-discursive (literally existing outside the text)' (Rudd 2020a, p. 98). To counter this notion, we might again slightly extend a partial quotation of Rose by Rudd: 'if children's fiction builds an image of the child inside the book, it does so in order to

secure the child who is outside the book, the one who does not come so easily within its grasp' (Rose 1984, p. 2). Rather than a non-ironic securing of the child of reality, I read in this a critique of the kind of argument repeatedly offered by Nodelman and Gubar and (heretically?) understood by Rudd as central to the work of Jacques Derrida:

> It doesn't take much reading of children's books to realize that the 'children' in the phrase 'children's literature' are not real human beings at all, but merely artificial constructs of writers; as is true of all works of literature, each children's story implies its audience; and thus each children's story reveals its author's assumptions about childhood. Rose is also right to insist on the limitations of those assumptions, and to demand our acknowledgement of them. In the often unconscious determination of writers to impose artificial ideas about childhood on their child readers, those writers do, often, fail real children—and anyone who cares for children or for children's literature should be conscious of that. (Nodelman 1985, p. 98)

A literature of revelation. Nodelman begins with an acknowledgement of 'artificiality', but this allows an author's assumptions seemingly to be secured. Likewise, and especially relevant to Rudd, from that same appeal to artificiality, the real child that artificiality fails can be secured. Without literature, I would suggest, claims to the real child become increasingly difficult. None of us, after all, dwell wholly within the real (that is what is so concerning about Nodelman's idea that care for real children should take place in the absence, or full knowledge, of projection). That is, we might say with a certain amount of cynicism, what is so *useful* about literature here: it allows precisely *a sense of artificiality against which the 'real' child can be 'grasped'*. I would, therefore, accept Rudd's claim that we 'only experience the Real indirectly, tangentially', (Rudd 2014, p. 46) but only with the caveat that we cannot remove this tangential experience from the Real.

Here I am reminded of the story of Zeuxis and Parrhasius, so beloved by Lacanians. The two painters compete to establish who is the greatest artist. Zeuxis paints grapes that look so real that birds fly down to eat them, and he thinks he has won, confidently pulling aside the curtain that conceals his rival's work, only to find his fingers fastening upon . . . paint. The curtain *is* the painting. If one wishes to offer up the real, so the story goes, one can expect a degree of praise for a work that acts as a lure to nature, but far better to set up an obstacle to vision. The literary child for Rose, we might say, is a *limit*. Through the inadequacy identified by Nodelman and Rudd, this child enables a notion of a real beyond it. This nothing behind the painted veil, Lacanians would claim, is far from irrelevant, a necessary condition for any symbolic operation. It is a nothing, however, that cannot stand on its own. No repression without return.[18]

## 5. A Challenge

I will conclude, as promised, with a return to the Lacanian text, although, significantly, and as indicated in the introduction to this article, the quotation I have chosen is not one called upon by tolerant and eclectic children's literature critics such as Rudd and Coats. I would suggest, therefore, that there is something intolerable for them in Lacan, with the Lacanianism they forward having many of its more troubling aspects foreclosed. The following is from *The Ethics of Psychoanalysis*, as translated by Dennis Porter:

> As long as it is a question of the good, there's no problem; our own and our neighbours are of the same material. Saint Martin shares his cloak, and a great deal is made of it. Yet it is after all a simple question of training; material is by its very nature made to be disposed of—it belongs to the other as much as it belongs to me. We are no doubt touching a primitive requirement in the need to be satisfied there, for the beggar is naked. But perhaps over and above that need to be clothed, he was begging for something else, namely, that Saint Martin either kill him or fuck him. In any encounter, there's a big difference in meaning between the response of philanthropy and that of love (Lacan [1986] 2007, p. 186).[19]

Let us begin with the standard response from Lacanian critics. This passage, describing the parable of St. Martin, who shares his cloak with a beggar, comes directly after a discussion of the strangeness of the Biblical injunction to 'Love they neighbour as thyself'. Freud saw this as an impossible demand (can we really invest the love we have for ourselves and our significant others in someone we do not know?), but Lacan sees a truth in these words: we fear our neighbour, rather than love them, to be sure, but the desire we see in them, destructive to us, reveals to us our own *jouissance*, that is, our desire for the Real object around which our desire circulates, an object that, if achieved, would rob us of the law that forms us as subjects. If we achieved *jouissance*, we would disappear as subjects. The end of our desire is our own destruction, and thus we have a terrible affinity with our neighbour: the *jouissance* I fear in my neighbour is that of the Real object—the nothing—that my desire circles around—that which is me more than myself. St Martin sharing his cloak remains at the level of the pleasure principle, an exchange of goods between interested parties. These goods—beliefs, material objects—are made up of the same material, because part of the symbolic order, and an interested party can exchange one good for the other, resulting in a basic satisfaction. What the beggar perhaps also begs is something else: beneath the demand for clothing is another demand, the *jouissance* that cuts across the pleasure principle, entails our destruction, and is the condition of true neighbourliness.

I have no reason at all to dispute this generally accepted reading of Lacan, other than the fact that I am not a Lacanian. What interests me is what *also* might be read in this account of what an individual also might have said. As I am nearing the end of this article, allow me to fix on a single issue: nakedness. In Dennis Porter's translation of Lacan, for example, 'the beggar is naked. But perhaps over and above that need to be clothed, he was begging for something else [ . . . ]'. What is 'over and above' nakedness has nothing to do with clothing, it is claimed, yet one might (heretically) read it to return as such *in being 'over and above'*. The begging to be fucked or killed is an addition, after all: 'something else'. In terms of this translation, we could counter Marc de Kesel's conclusion that "[b]ehind the ethical reproach 'I was naked and you didn't clothe me' thus lies an underside that is polymorphously-perverse as it is impossible" (De Kesel [2001] 2009, p. 148). It is rather that, for Porter's Lacan, this 'underside' is difficult to secure as wholly other than a clothing of nakedness. And it is not that nakedness shows itself fully; the idea of nakedness as final revelation or truth, the end of a process of unveiling that has entirely done with this. Rather, there is indeed 'something else' to it.

Nakedness, of course, always calls upon its opposite, ghosted as it is by the clothing with which it has seemingly dispensed. Nakedness cannot ever quite be itself alone.[20] With this in mind, I would contend that it is nakedness in this passage that constitutes the begging to be clothed, this because the beggar is not narrated as speaking: there is no initial begging to be read. It is, ironically, the 'something else' of the obscene supplement that constitutes the begging for the cloak. In one sense, therefore, it is not only the obscene begging that is 'over and above', but the straightforward economic begging for needs to be met. In terms of narrative perspective, the second begging is less the obscene 'over and above' of the explicit begging for clothing, than the begged yet unannounced underside of the begging to be killed or fucked. Thus, although in one sense it is clear that 'In any encounters, there's a big difference in meaning between the response of philanthropy and that of love', if we return to this text, and think about narrative perspective, then this 'encounter' is constructed, over and again, through uncanny repetitions, the non-Symbolic underside caught up in Symbolic deferral. To separate the two, we have to turn away from the text.

My interest, then, is with what Jacques Derrida describes as the '[t]he misrecognition or the failure to take account of the literary structure of narration, the omission of the frame, of the play of signatures, and notably of its parergonal effect' (Derrida 1988, p. 59). The passage discussed above, after all, and like the INTRODUCTION to *Winnie-the-Pooh*, includes a framing of speech by a third, with the beggar, for example, repeating his identity through his actions only within the perspective of the narration. In the terms Derrida uses

in the debate with the early, structuralist Lacan, footnoted in Rudd's *Lion and the Unicorn* article, the standard Lacanian reading that captures perfectly what is significant in the text does so only to 'denude' that text, and thus get safely to the point of danger: followers of Lacan claim that we can most clearly and radically see what is fundamentally other about us only if we ignore the deferrals and divisions in identity at the level of narration (Derrida 1988).

My argument, it should be clear, is not with the accuracy of the various readings offered by Lacanians such as Slavoj Žižek, Joan Copjec, Marc de Kesel, Tom Eyers, Alenka Zupančič, and Bruce Fink. We could not want better guides to Lacanian thought; they, of course, know far more about the subject than I ever will. Like Lacan in his account of St Martin, their interest is in what is and is not articulated—the obscene supplement to cosy exchanges in the Symbolic—this requiring the reader to risk a move beyond the Symbolic, and thus beyond specific textual formulations.[21]

There are powerful arguments against what I am forwarding here: I am sketching out a debate that the majority, if not all, Lacanian scholars have engaged. Thus, for example, de Kesel offers an exacting rebuke to my kind of thinking in *Eros and Ethics* when discussing precisely this passage (De Kesel [2001] 2009, p. 307), and Žižek, in the chapter 'Beyond Hegel' of his masterwork *Less Than Nothing*, where, it seems to me, he is at his most engaged with 'deconstructionist' arguments against his position, what I am suggesting here is recognised, but then subject to a succession of returns that work to take the ground from under it (Žižek 2013, pp. 480–504).[22] I will not stage these arguments here in detail: if you have not read them, they come highly recommended. Instead, I will close with a handful of statements or observations.

First, I will continue to argue for an approach to literature that, in its focus on textuality and perspective, finds itself at odds with Lacanian approaches. Second, my sense is that David Rudd might be more Lacanian than he realises. It is not, perhaps, in failing to engage the textuality of Lacan that he diverges from this school of thought. Instead, as a Lacanian, that might be considered his greatest achievement. Thirdly, I would argue that the difficulty for Rudd, again, in terms of his being a Lacanian, lies in his 'heretical' approach. A little orthodoxy is required, otherwise the 'framework' utilised cannot really be said to be owned by anyone. Lastly, my hope is that such orthodoxy would lead to a questioning of the politics of toleration—the tolerance of the tolerable—that he sees as the future of children's literature criticism. Lacanians may disagree on the ethics of obscene begging, but none recommend staying wholly within the economy of give and take. Lacan, I would suggest, asks us to go beyond the closed economy of children's literature criticism.

**Funding:** This research received no external funding.

**Institutional Review Board Statement:** Not applicable.

**Informed Consent Statement:** Not applicable.

**Data Availability Statement:** Not applicable.

**Conflicts of Interest:** The author declares no conflict of interest.

## Notes

1　This is why, for the most part, all three of us have moved on to different areas of research, although thinking about the child as construction remains at the heart of our practice. Sue Walsh researches the archives of the African Writers Series, Nigerian Literature, and Animal Studies; I research GRT identity, Žižekian psychoanalysis, and Critical University Studies; Karin Lesnik-Oberstein has been working on Wittgenstein, eco-criticism, and the failures of contemporary neuroscience. I should also add here, in terms of the understanding of the debate as parochial, that 'The Reading Critics' are not 'Anglo-Saxon', although given the name refers to our location in the UK, this is an understanable assumption. I am rare in the group in having English as my first language, for example. As indicated (below), those arguing against us similarly cannot easily be thought about as uniquely 'Anglo-Saxon' either.

2   I should point out here that the critics who oppose 'The Reading Critics' frequently disagree (see Rudd 2013, especially in terms of the suggestion that Nodelman does not do enough to acknowledge child agency), but the disagreement is seemingly enacted with tolerance.

3   I would like to extend my thanks to 'Reviewer 1', as I have borrowed a number of their formulations in editing this article.

4   For an extended justification of this position, see Lesnik-Oberstein (2016).

5   As evidenced above, Rudd's central target here is Kutzer, but I think the work of, say, AbdelRahim (2015) could also be understood in this way. I should add here that I think AbdelRahim offers a particularly astute reading of Milne's text. For further work on *Pooh* that tends to steer away from detailed reading, see Jacques (2015), Nance-Carroll (2014), Kidd (2011). I am not dismissing these texts (indeed, I would very much recommend them to anyone interested in the scholarly study of *Pooh*), but would suggest instead that their interest does not lie in textuality. I would especially direct readers unfamiliar with this scholarship to Kidd's historicising account of Milne's texts in relation to play and psychoanalysis. There is arguably a move to work against this tendency to avoid detailed close reading in the most recent volume dedicated to *Pooh* (Harrison 2021), which includes work by three of the critics mentioned above. Sarah E. Jackson's chapter on colonialism and Rudd's chapter on Derrida are particularly worth reading in this regard. I do, however, read in this text a repeated return of a non-textual 'reality' (through, for example, the psychological truth of cognitive dissonance, the notion of the 'childlike' or children's 'facility for detaching behaviour from its consequences', or the actuality of animals). Rudd's chapter, it should be said, does not fall into the more obviously problematic understanding of deconstruction forwarded in his article for *The Lion and the Unicorn*, where Derrida is figured as some kind of referential thinker (as discussed in this present article). Rudd compares his approach in Harrison's text to the kind of deconstructive response to *Pooh* satirised by Frederick Crewes: Rudd's is 'a more straightforward explanation of how the Pooh books chime with some of Derrida's key ideas. I do hope that readers appreciate the "différance"' (Rudd 2021, p. 3). If I were to address this more recent text at length, my starting point might be how 'key ideas', 'explanation' and 'chim[ing]' might be read in terms of Derrida's texts. Among the exceptions to the neglect of the text in *Pooh* studies, one that has significantly not been referenced in any subsequent critical account of Milne, and that I build on in this article, is Lesnik-Oberstein (1999).

6   See Jackson (2021) for an additional critique of this kind of 'political' argument. See Cocks (2022) for an example of a critique of this kind of non-textual approach to emancipatory and identity politics.

7   This argument is repeated in Rudd (2021).

8   I am thinking here, for example, of Jessica Lynne Pearson's suggestion that 'the system of colonial oversight evolved in an ad hoc manner' (Pearson 2017). See Joronen (2019) for a recent account of the interweaving of the ad hoc and the planned within a contemporary colonial setting. In terms of literary study, probably the most well known, related account of the colonial dimension of play and improvisation is by Greenblatt (2005). In terms of a more general critique of play as an expression of power, see Mould (2018) or Fleming (2014).

9   For Rudd, in the glossary that accompanies his 2013 text, the Symbolc is 'the order of language (though language is not unique to this order) [...] an order predicasted not on things but the way they are organised/structured [...] a realm of signifiers, where nothing has a positive value [...] it is also the realm of the law, of our culture, and, in accepting it, we are allocated a place [...] as with the mirror image of the Imaginary, we come to identify with something outside ourselves, which grants us meaning' (Rudd 2013). I have no issue with this or any of the other definitions Rudd offers in the glossary, other than that they leave open the question of how they relate to the approach he takes in the main body of his text.

10  I have included this possibility at the suggestion of one very detailed, anonymous reading of an initial draft of this article, prior to submission to *Humanities*. The notion is that in 'the letter of the text' Rudd is appealing to the kind of reading of 'the materiality of the signifier' forwarded by Fink (2004) and Žižek. Despite both theorists being referenced with approval by Rudd, I do not see this to be the case in this quotation, as explained above, but it is an interesting suggestion.

11  This difference is not simply a matter of intention, of course. There are constraints within language that impact upon certain formulations: it would not make sense here if Christopher Robin 'used to call a swan Pooh', for example.

12  An alternative reading might move from the sense in which 'we didn't think' signifies a lack of thought. Might this actually be a knowledge claim to thought? That would certainly impact on my reading here. Thank you to Reviewer 2 for pointing out this ambiguity.

13  The Lacanian letter is referenced by Rudd, but it is a point of contention, as indicated previously and as I will tentatively suggest in the conclusion to this article. Again, are we to read this 'letter' in terms of a broad notion of fidelity to the text? Or as a specific reference to the materiality of the signifier, of the kind forwarded by Fink?

14  'Surplus pleasure' is precisely it, of course, but, as I read in this present article, I am not convinced that Rudd makes an adequate case for why this is so. Rudd does offer a more sustained and subtle reading of *jouissance* in a later article (Rudd 2020b), but here still I would argue that a number of the problems described in this present article remain.

15  Rudd, it should be remembered, does elsewhere offer readings within what might be termed a broadly Žižekian framework (Rudd 2013, 2020b).

16  The footnote on p. 199 of Rudd (2013) discussing the nothing of the phallic is only a limited help here. It can be problematically read in relation to the 'obviously' phallic imagery of Hook's pipe (again, p. 199), for example. See Rudd (2020b) for additional examples. I should add here that, to my mind, the majority of engagements with Lacan within children's literature criticism

follow what I am reading as an approach based on the notion of the literary text as an allegory for psychoanalysis. See especially Coats (2007). For exacting critiques of Coats' allegorical and therapeutic understanding of the Lacanian project in relation to children's literature, see Lesnik-Oberstein (2016) and Buckley (2018). Rudd himself argues against this allegorical approach in a review of another work on Children's Literature and Psychoanalysis (2001). That said, I think Rudd might find support from Žižekians both for his notion of literary texts demonstrating psychoanalytic principles, and his appeal to symbolism (see 'Hitchcock's Sinthomes' from Žižek 1992, for example).

[17]　The Real, for Rudd at this stage, is, as I understand it, the location of Ideological State Apparatus, and he regrets Rose seemingly not having the understanding of Marxism that would have allowed her to acknowledge this (Rudd 2020a, p. 94). Here I would, with a matter of urgency, direct Rudd's attention to the chapter of Rose (1984), 'Peter Pan, Language and the State: Captain Hook Goes to Eton'. This really is an extraordinary omission on Rudd's part. Quite simply, I take Rose to be a *political* writer.

[18]　Yes, Rudd does write of real meanings not being behind a door (Rudd 2014), but the Real is not real meanings, and, in any case, in my understanding occasional references to recognisable Lacanian formulations point to an inability to carry through a reading as much as moments of salvation.

[19]　I have chosen the English translation of Lacan by Dennis Porter, as it is widely seen as a standard translation, utilised also in the English translation of de Kesel. Crucially, in 'returning to Lacan', I am following here Rudd's general recourse to English translations of French theory. It is clear that Rudd assumes his reader will be engaging theorists in English, and that the 'nimble' or 'dynamic' response he champions will be to an English text. He claims, for example, that Sue Walsh's seeming inability to gain his own nuanced understanding of Derrida's 'nuanced approach' is excusable because: 'To be fair, Derrida's essay was available only in French at the time Walsh's work was published' (Rudd 2020a, p. 103). That Walsh is bilingual is seemingly unthinkable to Rudd. It is worth pointing out here that reading Derrida and Lacan in French and Freud in German is not a startling activity for 'The Reading Critics'. The work of our Graduate Centre at Reading has always been international, and Comparative Literature is one of our many specialisms. It is important to stress, therefore, that to return to the original French of Lacan would open up new readings, and invalidate, or at least question, those that I have offered. My reading here is precisely of the translation, in other words. If one were to offer a detailed comparative analysis, the most obvious starting point would be the lack of a 'neighbour' in Lacan's account: 'Tant qu'il s'agit du bien il n'y a pas de problème, parce que ce qu'on appelle le bien, le nôtre, et celui de l'autre, ils sont de la même étoffe'. At this point, there is none of the play of 'le voisin' and 'le prochain' that Dany Nobus reads in his account of translation and 'Kant avec Sade' (Nobus 2017, pp. 130–33). Instead, there is a consistent construction of 'the other' across the passage. Even more problematic, for my reading, are the formulations around the need for nakedness: 'Le mendiant est nu, mais peut-être au-delà de ce besoin de se vêtir mendiait-il autre chose, que saint Martin le tue, ou le baise.' This is not, then, a question of what is 'over and above' needs, but what is beyond, or further to, them: 'au-delà de ce besoin'. In other words, we might start thinking in terms of 'Au-delà du principe de plaisir', rather than the precise language of clothing and nakedness familiar from 'Le séminaire sur "La Lettre volée"'. What is especially interesting to me here, and one possible direction a future reading could take, is the sense in which the translation allows us to retroactively read and refigure the original, an operation that could, for example, be approached through the dialectics of Žižek's 'The Limits of Hegel'.

[20]　This reading is drawing on Jessica Medhurst (2019). Shortly before going to press, I also became aware of the recent, essay on nakeness, clothing and Lacan by Peter D. Mathews (2021), which I strongly recommend.

[21]　For this, see especially Copjec (2015).

[22]　More obvious, perhaps, is the trenchant critique in Žižek (2001) of what Žižek sees as a certain kind of theorist who praises Judith Butler, Jacques Derrida, and text-based criticism, whilst regarding the Holocaust as an evil only to be met with silence. I recognise myself in this, I have to admit, however much I would claim this also as a misrecognition on Žižek's part. For an account of Lacanian thought that would be critical of my own, but also would both qualify issues of the Symbolic and the Real in the work of the critics named above, and point to the difficulty of Rudd claiming his 'framework' to be in any sense 'Lacanian', see Eyers (2012). For my own critiques of Žižekian Lacanianism, see Cocks (2015, 2020a, 2020b, 2021, 2023).

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
