# Peer review of "On the Tolerance of Children’s Literature Criticism: Psychoanalysis, Neighborliness, and Pooh"

_humanities, doi:10.3390/h12030045_

Round 1

Reviewer 1 Report

See attached file for comments and suggestions

Author Response

Thank you so much for this exacting and encouraging review. Within it, you articulate exactly my aims in writing this article:

"[it] will be to the benefit of children’s Literature and Childhood Studies more widely. The article represents a novel and useful contribution not only to these two areas, but also to the philosophy of tolerance, and (despite the author’s protestations) Lacanian studies. It aligns with Humanities interdisciplinary interrogation of human existence and its scope of cultural and literature studies, and offers a remarkably novel understanding of an established debate."

I agree with all you suggestions for minor revisions, and have made appropriate changes.

  • On the advice of Reviewer 2, I have rewritten the Introduction (I found this to be really productive advice, as the rewrite clarifies certain ideas). An explanation of the initial reference to ‘neighbourliness’ now appears within this introductory section, rather than emerging only at the articles end.
  • I have adjusted the pronouns to make them consistent (I not we etc).
  • I have adjusted the tone. Very much agree with your comments here. The ‘o boy’s etc have been removed.

I hope this is OK, but I have also utilised a number of your endorsing formulations in my new introduction, to further explain the purpose/aims of the article. I have acknowledged that these are your formulations in a footnote.

Reviewer 2 Report

The article is a reply to a paper by David Rudd that criticizes the author's and his colleagues' work, reproaching them to engage ‘in a particular futile form of theory war’. As an answer, the author critiques Rudd’s Lacanian approach to Winnie-the-Pooh in his work Reading the Child in Children’s Literature. An heretical approach and shows the inner contradiction of Rudd’s celebration of ‘dynamic’ reading – a reading that is liberated from narrow interpretative frameworks and that seemingly strives towards ‘jouissance’. After a short introduction, the author agrees with Rudd on the fact that current political/ideological approaches to children’s literature neglect the text, and illustrates this by showing the flaws of a political reading of the introduction of Winnie-the-Pooh. The author then points out that Rudd himself doesn’t honor his own position that demands an attentive reading. David Rudd uses Winnie-the-Pooh to contradict Jacqueline Rose’s claim about the transparency of language in children’s books and to show that Milne isn’t ‘colonizing’ his young readers, as claimed by other scholars. The author shows how this argument is based on a non-attentive reading of the text, on a false understanding of colonialism and on the (debatable) idea that one can make a stake about the author’s state of mind (‘tentative and uneasy’) through reading a text.  In section 4 (section 3 is missing) the author attacks Rudd presupposition that a reading that follows no pre-definded interpretative framework releases the ‘jouissance’ of a text. The author claims that Rudd’s idea that the jouissance of reading comes from the absence of a repressing framework ignores the Lacanian theory that jouissance needs repression in order to deploy itself. The author also discuss the Lacanian understanding of jouissance and desire. They end the section by questioning whether Rudd’s theoretical framework is really Lacanian since his use of Lacanian vocabulary doesn’t take into account all the aspects of the Lacanian thought. In section 5, the author continues pointing out Rudd’s lacking framework. They link Rudd’s loose way of applying Lacanian psychoanalysis to the texts to Rudd’s self-claimed ‘heretical’ approach. The author reproaches Rudd that his choice of Lacanian framework is ‘rooted in a whim’. They also reproach Rudd his over-confidence, as he applies the Lacanian framework very loosely himself, but still  dares to ‘’suggest(s) to Jacqueline Rose, arguably the UKs leading Lacanian scholar, how she might up her game”. They furthermore question Rudd’s understanding of literature and his understanding of the difference between Jacqueline Rose and the Reading Critics – a group of critics the author is part of. In their concluding section, the author discusses the limits of a largely accepted interpretation of an extract of Lacan’s Ethics of Psychoanalysis (that isn’t linked to Rudd or to children’s literature). They conclude their paper by stating that they will always be in favor of an approach to children’s literature that focuses on textuality (and is thus in opposition with the Lacanian approach) and by commenting on Rudd’s relation to his Lacanian framework.

The paper is an important critique of David Rudd’s Reading the Child in Children’s Literature – a book that has been generally well received by the community of children’s literature scholars. Nevertheless,

·       -it needs more contextualization in order to also make it understandable and relevant for people who aren’t specialists of Lacan, of psychoanalytic approaches to children’s literature or who don’t have an insight into the ‘war of theory’ going on between (Anglo-Saxon) children’s literature scholars. èPlease explain Lacanian concepts and positions like e.g. his resistance to interpretation, the uncanny psychoanalysis, the Lacanian letter, fundamental fantasy, the Symbolic,… èand give an insight into the debate around the child in children’s literature to give some context to readers who don’t come from Children’s and YA literature or aren’t aware of the Anglo-Saxon debates around the child in children’s literature.

·      it lacks a section that explains the framework. People outside of children's literature studies and non anglo-saxon children's literature scholars are likely to ignore who Rudd is, who the Reading critics are, how Rudd’s work is received in the community (no other paper on Rudd’s work is mentioned)…

·       Right now, the points on which you attack Rudd sometimes seem a little arbitrary. Why did you choose Rudd’s analysis of Winnie-the-Pooh and not Rudd’s reading of another work? Why choose to focus on Rudd’s (mis)understanding of ‘jouissance’ when you also say that you see many more problems? Why conclude with a section on an extract of Lacan that has not been discussed by Rudd (or by any children’s literature scholars)? è  It would help if in the beginning of the paper, there was a section on methodology and presentation of the different works of Rudd that will be analyzed.

·       Sometimes it is difficult to follow the rationale, as there seem to be ellipses in the formulation of it (see comments on single extracts on the page below).

·       In the introduction and in the abstract, the article emphasizes on the question of tolerance, of violence and erasure, but doesn’t come back to those questions in their analysis. è Please either take this announcements out or engage with these announced topics.

·       I personally felt that the concluding section discussing current interpretation of an extract of Lacan’s Ethics of Psychoanalysis (that has not been discussed by Rudd) mades the paper move away from its initial point that focused on Rudd. èI think the analysis of Lacan could be shortened and the link to the rest of the paper could be made more prominent by putting it in the beginning of the section and developing it.

·       I’m wondering whether announcing the article as a reply to Rudd’s critic of your work is the best way to go. Since you don’t directly reply to the paper that criticizes your work, but dissect other work of Rudd, I think that ‘making it personal’ weakens your stance, as it might seem as an attack that is motivated by personal issues rather than by scientific ones (even if by reading the paper it becomes clear that you have some solid points). è to avoid this, I suggest that a) you add another motivation for the writing of this article than Rudd’s attack on your work (show why your article is relevant to people who aren’t involved in the quarrel) and è b) you show more clearly how the problematic aspects Rudd’s work is related to his critique towards your work. (How is it a ‘reply’?)

·       I’m not sure how this article fits in an issue on ‘Constructing the Political in Children’s Literature’. Is it supposed to be published in the Varia-section? If not, è I suggest you stress out it link to the topic of the issue.

7: What do you mean by ‘close reading of…psychoanalysis’? Please clarify or rephrase.

48: Erasure of whom/what?

73-95: Does Rudd comment on AbdelRahim’s reading of the INTRODUCTION? If yes, please mention it. If not, I’m not sure these lines are necessary since they distract the reader from your critique on Rudd’s work.

118: Please explain what you mean by ‘political good’.

119-125: I don’t quite understand what point you are making here. Could you please clarify?

208-2011: Please clarify how the call to pay 'attention to the letter of the text' is an 'acceptance of otherness ... that is not other.'

233-243: Your argument is enlightening, but it would sound very strange if Milne had written ‘he once had the swan’… Is there a way to present this argument taking into account the limits of the creational freedom of authors? (i.e. taking into account that if Milne’s had made the relation between Christopher and the swan symmetrical by saying ‘the’ swan, it would have sounded like bad English…?)

244: Please clarify in what sense the lack of knowledge on the part of the narration did initially signify generosity or tolerance towards the other.

246: I can’t follow your rationale when you say ‘Because it is known that we were thinking that we did not know what this other was thinking, it can be divested of the object that is its name’. To me the sentence you are referring to ( ‘we didn’t think the swan would want it any more’), sounds like they think they know what the other wants (or doesn’t want). Please enlighten me here.

254: Replace ‘4’ by ‘3’ (and adjust following numbers)

257: Why the italics in ‘Winnie-the-Pooh as a reading of Lacanà is that a quote? A title? Please clarify. In which work does Rudd do a reading of Lacan in Winnie-the-Pooh? And please explain the difference between a Lacanian reading of Winnie-the-Pooh and Winnie-the-Pooh as a reading of Lacan.

277: ‘fundamental fantasy’àplease indicate that this is a Lacanian concept and since you [put it into quotation mark, I also suggest you put the exact source.

565-566: . ‘It is not, perhaps, in failing to engage the textuality of Lacan that he diverges from this school of thought’à I don’t understand that sentence  (maybe it is because I’m not a native speaker). Could you please rephrase?

566-567: Please clarify.

Author Response

Thank you for such a sensitive and precise review.

At the end of the review, there are a list of specific suggested changes. I have positively addressed all but two of these.  As instructed by the Humanities Journal, I will begin this 'Author's Reply' by listing these, and indicating how I have acted on them, and where I haven’t made a change, offering an explanation.

I then address the numerous broader suggestions you make. The majority of these I have acted on positively. For example, I have added a section on methodology at the start, indicated why the debate matters for those not within it, clarified the texts by Rudd I am talking about, set out in advance my argument, deleted the section on AbdelRahim, made clear why I need to read Lacan at length at the end etc. On a few points I am suggesting there should not be a change, such as the reading of Lacan in the closing section of the piece, but here I will offer a justification/explain my thinking (but, essentially, I think the reframing of the article you suggested helps explain what the ending is doing. Without that change, I realise that the ending would seem obscure. In other words, the lack of a change is only possible because of others changes I have enacted through following your suggestions.)  

Just to add here, that initially I was concerned about making all the changes, but once I had, I realised I really was wrong to be worried: they clearly have a positive and transformative effect.  This is, I think, a much improved article thanks to your productive and insightful suggestions…

  1. Specific changes

7: What do you mean by ‘close reading of…psychoanalysis’? Please clarify or rephrase.

I have substantially rewritten/rephrased this section, and it is hopefully all now clearer.

48: Erasure of whom/what?

Because I have also added a justification for those not involved with the debate for staging an encounter between Rudd and The Reading Critics, the question of who is being erased is now hopefully clearer, and there is more of a sense of it being a sustained concern across the article.

In one sense, in this quotation I am simply referring to Rudd’s seemingly generous dealing with The Reading Critics, but suggesting that his tolerant attitude could be read as a form of aggression or a move to close us down and render us insignificant – the idea that our work is sterile and surpassed that comes from the discussion of 'a particular futile form of theory war'.  The ‘we’ in the sentence you indicate refers to The Reading Critics, as set up in the previous sentence. The form of the question ‘can we read erasure in it’ is deliberately open. I am asking ‘What might be the violence/erasure of tolerance?’ It is this question that is then engaged through the reading of Pooh and colonialism, and is central to the final section on Lacan, where Rudd’s ‘Lacanian framework’ is framed in turn through a reading of an extract from Lacan that would never make it into discussions with Anglo-Saxon/liberal humanist children’s literature criticism, an extract that also suggests that neighbourliness might contain in it counter-intuitive, violent, and self-destructive desires.

In short, I have substantially rewritten the introduction to make this point about erasure clearer earlier on.

73-95: Does Rudd comment on AbdelRahim’s reading of the INTRODUCTION? If yes, please mention it. If not, I’m not sure these lines are necessary since they distract the reader from your critique on Rudd’s work.

Good point, thank you. I have deleted this section. To clarify my initial point, I was using AbdelRahim to kind of ‘be nice’ to Rudd. For me, AbdelRahim is great, but her approach suggests that Rudd also has a great point in what he is arguing. I was suggesting that Rudd’s Reading the Child has real validity, because we can extend his critique of politic responses to children’s literature that do not engage in close reading beyond those few texts on colonialism and narration he does discuss. His commentary is relevant even to texts, such as AbdeRahim’s, published after his own and not simply working within an Anglo-Saxon frame. But, exactly as you write, this section really does end up as a distraction: its not needed.  

118: Please explain what you mean by ‘political good’.

Agreed, this was unclear. Have reformulated both references to this.

119-125: I don’t quite understand what point you are making here. Could you please clarify?

Yes, I have added a second formulation to clarify.

208-2011: Please clarify how the call to pay 'attention to the letter of the text' is an 'acceptance of otherness ... that is not other.'

Yes, I have reformulated. My reference was to the ‘arising’ and that was not clear in the initial formulation.

233-243: Your argument is enlightening, but it would sound very strange if Milne had written ‘he once had the swan’… Is there a way to present this argument taking into account the limits of the creational freedom of authors? (i.e. taking into account that if Milne’s had made the relation between Christopher and the swan symmetrical by saying ‘the’ swan, it would have sounded like bad English…?)

Agreed. Have added a note that underlines the point you make here.

244: Please clarify in what sense the lack of knowledge on the part of the narration did initially signify generosity or tolerance towards the other.

I have added a sentence that clarifies this, have adjusted the initial formulation to further clarify, and there are further additions also that aim to carry this point through.

246: I can’t follow your rationale when you say ‘Because it is known that we were thinking that we did not know what this other was thinking, it can be divested of the object that is its name’. To me the sentence you are referring to ( ‘we didn’t think the swan would want it any more’), sounds like they think they know what the other wants (or doesn’t want). Please enlighten me here.

Great point. Im reading ‘we didn’t think’ in the colloquial sense that I would describe as something like ‘negativity’. In other words, ‘we didn’t think’ reads to me as a lack of thought. It describes thoughtlessness.  But it works the other way round as well, as you demonstrate. I’ve noted this. 

254: Replace ‘4’ by ‘3’ (and adjust following numbers)

Yes!

257: Why the italics in ‘Winnie-the-Pooh as a reading of Lacan’ à is that a quote? A title? Please clarify. In which work does Rudd do a reading of Lacan in Winnie-the-Pooh? And please explain the difference between a Lacanian reading of Winnie-the-Pooh and Winnie-the-Pooh as a reading of Lacan.

Have deleted italics (they were there for ‘emphasis’ – agreed, it didn’t work…). The difference between the two formulations is that the first indicates to me a reading of Pooh using a Lacanian Framework, whereas the second suggests that the reading of Pooh produces a certain idea of Lacan. That is to say, within the first formulation we kind of take it as read that there is something called ‘Lacan’ that we can apply to a primary text, whereas in the second the reading of the primary text gives us a ‘version’ of Lacan. I have added ‘and commentary on Lacan’, which I hope helps to clarify the point.

277: ‘fundamental fantasy’ please indicate that this is a Lacanian concept and since you put it into quotation mark, I also suggest you put the exact source.

Agreed, have added reference.

565-566: . ‘It is not, perhaps, in failing to engage the textuality of Lacan that he diverges from this school of thought’. I don’t understand that sentence  (maybe it is because I’m not a native speaker). Could you please rephrase?

I do like this sentence, and would like to keep it. In a really straightforward way, I am saying that a) Rudd thinks Lacan is all about close reading b) But I think that’s only true in a very specific sense. As indicated, Lacanians tend, for complex reasons of their own, to resist close reading. But Rudd really has no idea that they do this, because he really has little engagement with either what Lacan actually writes or with contemporary debates amongst Lacanians. C) Rudd prides himself on his close reading D) But he tends not to close read theory, including Lacan (and, as I have indicated, the close reading of primary texts he engages is often problematic) E) Rudd thinks he is ‘broadly Lacanian’ because he close reads. F) I think, ironically, he is ‘broadly Lacanian’ because he doesn’t!

I think the sentence works in cautiously and ironically setting this up. Ive had a go at reformulating, but I do think all my versions lack the elegance and economy of this present formulation. There is a 'rhythm' to this final paragraph that I really like, and would not like to see lost. 

To clarify: ‘this school of thought’ refers to the Lacanianism of the previous sentence.  

566-567: Please clarify.

Again, I think this one works – to clarify, I am saying (as above) that Rudd thinks Lacanian’s love close reading in a traditional sense, whereas in fact they generally don’t. Or, one might say, their close reading is tied to a move beyond the symbolic, and isn’t of the kind Rudd would recognise, as indicated in my reading of St Martin. I am also saying that Rudd doesn’t really do close reading in terms of theory. So in failing to close read, Rudd is actually quite Lacanian. I think over elaboration here would work against the economy of the point. As suggested above, there is an economy, precision, 'rhythm' and also 'wit'  to this final paragraph that I would not like to see lost. Meaning and rhetorical force would be lost if I reworded.  

  1. General changes
  • -it needs more contextualization in order to also make it understandable and relevant for people who aren’t specialists of Lacan, of psychoanalytic approaches to children’s literature or who don’t have an insight into the ‘war of theory’ going on between (Anglo-Saxon) children’s literature scholars.

Agreed. I have rewritten and extended the entire first section in light of this. In doing so, I called upon formulations from ‘Reviewer 1’ of this article. In praising the article, they actually formulated for me what it was doing in a way I now realise I should have done myself. I acknowledge the debt in an endnote.

One important side-point, however: although it is important to justify why a general audience should be interested in what can be understood to be a parochial debate, ‘The Reading Scholars’ are not Anglo-Saxon. Because we are named after the place at which some of us work, which is in the UK, this is understandable. I am, however, the only one of the initial group whose first language is English, and I can only think of two other scholars in the extended network of published academics aligned with the group who could be considered Anglo-Saxon. We are Cuban, Dutch, Italian, Korean, Chinese, French. The Graduate Centre in which we are based  in the UK(CIRCL) has equivalent organisations in Malaysia, Korea, China, and Japan. 

Please explain Lacanian concepts and positions like e.g. his resistance to interpretation, the uncanny psychoanalysis, the Lacanian letter, fundamental fantasy, the Symbolic,…

I have added explanations, some in the main text, some in footnotes. I do not want to place too much emphasis on these explanations, however, for a number of reasons. The first is that Rudd’s criticism does this, and in a way I find problematic. What we get are brief explanations, designed for a non-specialist audience. So, jouissance = x and y. The extended section on jouissance in my article indicates why this kind of approach is problematic – one really needs to work through these ideas at some length to get any idea of how they are functioning. Lacan is challenging, and an explanation of ‘the Lacanian letter’ or ‘symbolic’, to offer any real help, would have to be as at least as long as the section on joussiance (as I now explain in detail at the start of the article, jouissance is chosen because it sets up neighbourliness and questions of tolerance and community, which is what links the introduction and the conclusion to the rest of my article). Some are not as complex as they seem: the reference to ‘uncanny psychoanalysis’ is a rhetorical flourish on my part rather than a technical term, for example, and I have made a slight adjustment to clarify this. ‘Resistance’ can be understood in terms of its conventional meaning (I could go on to explain the complexity of the psychoanalytic implications of the term via Derrida's Resistances of Psychoanalysis, for example, but that would over-extend this article – a reader will be able to follow my argument by reading the term in a basic way).  ‘Fundamental fantasy’ and ‘the Symbolic’ do need an explanation or reference – agreed. I have looked to Rudd for a definition of the symbolic, because what he writes about it in the glossary to his book is great, but also because this allows me to suggest the inadequacy of such formulations within a glossary. I don’t want to be offering a simplified Lacanian framework that ends up being not very Lacanian at all.  For me, it is significant that successful dictionaries of this kind (I am thinking here especially of Butler’s Zizek Dictionary) resist simple explanations, and opt instead for discursive short essays from multiple authors with different interpretations. The glossary definition gives a false sense of mastery, and this is, I think, at the heart of the problem of Rudd’s Lacanianism. But hopefully the adjustments I have made give enough context to make the article work.

 it lacks a section that explains the framework. People outside of children's literature studies and non anglo-saxon children's literature scholars are likely to ignore who Rudd is, who the Reading critics are, how Rudd’s work is received in the community (no other paper on Rudd’s work is mentioned)…

Agreed. Again, have substantially rewritten the first section to address this. There has been a major revision here, and I think the result is a much improved and far clearer article.  

 Right now, the points on which you attack Rudd sometimes seem a little arbitrary. Why did you choose Rudd’s analysis of Winnie-the-Pooh and not Rudd’s reading of another work? Why choose to focus on Rudd’s (mis)understanding of ‘jouissance’ when you also say that you see many more problems?

Agreed. Have now addressed this at length. Again, the result is, I think, a much improved and clearer article.  

It would help if in the beginning of the paper, there was a section on methodology and presentation of the different works of Rudd that will be analyzed.

Yes! Really agree. I have reshaped the article around this suggestion, and have significantly rewritten. I hope this is now clearer.

Why conclude with a section on an extract of Lacan that has not been discussed by Rudd (or by any children’s literature scholars)?

The final section on Lacan is essential to my argument, but I can see that this was not clear in the first draft. It is actually precisely the neglect of passages such as this in children’s literature criticism that is of interest to me.

At the start of the article, I have added an explanation of why I am reading the extract from Lacan at the end. Essentially: 1) it does what Rudd never does, indeed, what virtually no children’s literature critic does, and stages a detailed reading of Lacan. My suggestion is that the very act of critically reading Lacan’s words in a way that thinks about framing and textuality is path-breaking in the field of children’s literature criticism. 2)  I deliberately chose an extract from Lacan that no children’s literature critic has or ever would read. Such critics not only do not read Lacan in detail, and generally stay with well known Lacanian texts (most obviously The Mirror Stage), that can be redeemed in some way – a cosy Lacan. But so much of Lacan is just shocking! My point here is to ask ‘what kind of Lacan is Children’s Literature Criticism reliant upon if it does not engage ethically difficult passages such as this?’ 3) The parable of St Martin is chosen because it is about neighbourliness. Rudd’s children’s literature criticism , as shared by so many of the major figures in Anglo children's literature criticism, is concerned with tolerance, and the promotion of a community of debate. But the force of Lacan’s critique here involves a witty/horrible critique of such tolerance: essentially we are neighbourly because we are nothing/we wish to annihilate ourselves/because of drives that undermine all the sense and meaning community is founded upon. The politics I am critiquing in this article is this fundamental politics of tolerance – or the tolerance of the tolerable, as I call it. For Rudd, everyone should just get along and accept the differences of others. Lacan, for me, draws out was is so problematic about that ‘liberal’ logic, the violence and erasure that fuels it, but also the violence and erasure that it entails. And thus, from many angles, the St Martin piece can be returned as a critique of the whole project of Anglo-Saxon/liberal human children’s literature criticism.

Sometimes it is difficult to follow the rationale, as there seem to be ellipses in the formulation of it (see comments on single extracts on the page below).

As indicated above, I have addressed this.

In the introduction and in the abstract, the article emphasizes on the question of tolerance, of violence and erasure, but doesn’t come back to those questions in their analysis. Please either take this announcements out or engage with these announced topics.

See my comment above.

The ideas if tolerance, violence, and erasure are what the whole article turns on. The idea that the Reading Critics are no longer relevant; the idea that our apparent intolerance should not be tolerated; the celebration of an approach to theory that is tolerant of difference and is eclectic; the idea that such tolerance and eclecticism can be read in Pooh; how this tolerance involves a broad approach to theory which leads to an erasure of Lacanian ideas, a neglect of detailed reading of precise Lacanian language,  and an erasure too for crucial aspects of quotes from Jacqueline Rose; how this question of tolerance is related to a failure to read the repression necessary to jouissance, and how this relates in turn to an idea of a knowable child that is seemingly knowable in a way that is free from restrictive frames, just as the jouissance of the text is free from the ‘monologic of the symbolic’.   

I have kept the ending, but hopefully, now recontextualised by an explanatory introduction, and through changes throughout the article, it is now clearer how it fits within the argument as a whole.

 I personally felt that the concluding section discussing current interpretation of an extract of Lacan’s Ethics of Psychoanalysis (that has not been discussed by Rudd) mades the paper move away from its initial point that focused on Rudd. I think the analysis of Lacan could be shortened and the link to the rest of the paper could be made more prominent by putting it in the beginning of the section and developing it.

See my answer directly above for justification for this.

I ’m wondering whether announcing the article as a reply to Rudd’s critic of your work is the best way to go. Since you don’t directly reply to the paper that criticizes your work, but dissect other work of Rudd, I think that ‘making it personal’ weakens your stance, as it might seem as an attack that is motivated by personal issues rather than by scientific ones (even if by reading the paper it becomes clear that you have some solid points). To avoid this, I suggest that a) you add another motivation for the writing of this article than Rudd’s attack on your work (show why your article is relevant to people who aren’t involved in the quarrel) and b) you show more clearly how the problematic aspects Rudd’s work is related to his critique towards your work. (How is it a ‘reply’?)

I really do recognise the force of this critique, but I think that my approach is a nonetheless appropriate. Rudd has questioned my work, as have his colleagues, and theirs is a very powerful voice in children’s literature criticism. This is my reply. It is an oblique reply because if I were to address only his points in his article, I would not only be perpetuating what is an interminable confrontation, but working within a frame of response he has set out. This article is about shifting the debate. Shifting it, indeed, so it isn’t a debate. Instead, it asks: what might a Lacanian approach to Childrens Literature involve? What kind of knowledge and approaches would be needed? What in Rudd’s work in general works again claims to be working in a Lacanian way? How are claims to work with a plurality of approaches undermined by a lack of consistency (it is worth mentioning here also that this was the one of the aspects of the article that Reviewer 1 singled out for special praise – for them, the way I approach Rudd, and reframe the debate, is one of the great justifications for publication) 

I should add also that I do end up ‘directly reply[ing] to the paper that criticizes [my] work’ in the penultimate section, but I can only do this after the shift my readings of Rudd on Pooh and psychoanalysis, and a wider reading of jouissance, has enabled.  

Significantly, however, Reviewer 1 also noted something ‘personal’, but located it firmly  (and correctly, I think, I’m ashamed to say…)  in the rather passive aggressive voice I sometimes adopt. I have corrected this throughout, and hopefully that works also to lessen the sense of a personal attack

 I’m not sure how this article fits in an issue on ‘Constructing the Political in Children’s Literature’. Is it supposed to be published in the Varia-section? If not, I suggest you stress out it link to the topic of the issue.

The politic is in the fundamental appeal to tolerance and plurality, but I have rewritten to emphasise this more. It is a concern, as I argue above, that is engaged through the whole article. Rudd sees explicitly ‘political’ critics of children’s literature as reductive. His is a politics of community, eclecticism and tolerance: why can’t we all just get along? My purpose in this article is not only to draw out what is problematic in general about such tolerance, but to suggest the particular difficulties that arise from a politics of tolerance rooted in a broad approach to theory, and also, fundamentally, in Lacan.  The article supports the a number of the Special Issues research questions, most significantly, I think:  

  • What authority or law is required within narratives of rebellion or resistance?  What are the limits of subversion? 

It also engages with the question of how textual framing impacts on the notion of the political, and also advocates the close reading approach that the CFP sees as an approach of special interest. 

Round 2

Reviewer 2 Report

Thank you for addressing my concerns in such detailed way. The paper is much clearer now!